# A *Bacillus subtilis* Strain ZJ20 with AFB1 Detoxification Ability: A Comprehensive Analysis

**DOI:** 10.3390/biology12091195

**Published:** 2023-08-31

**Authors:** Meixue Huang, Jing Guo, Yanyan Jia, Chengshui Liao, Lei He, Jing Li, Ying Wei, Songbiao Chen, Jian Chen, Ke Shang, Rongxian Guo, Ke Ding, Zuhua Yu

**Affiliations:** 1Luoyang Key Laboratory of Live Carrier Biomaterial and Animal Disease Prevention and Control, College of Animal Science and Technology, Henan University of Science and Technology, Luoyang 471023, China; meixue48264@163.com (M.H.); 18317545493@163.com (J.G.); jiayanyan0120@163.com (Y.J.); liaochengshui33@163.com (C.L.); helei4280546@163.com (L.H.); lijing5221@126.com (J.L.); 9906127@haust.edu.cn (Y.W.); chensongbiao@126.com (S.C.); chenillejian@163.com (J.C.); shangke0624@163.com (K.S.); guorongxian520@163.com (R.G.); 2Laboratory of Functional Microbiology and Animal Health, College of Animal Science and Technology, Henan University of Science and Technology, Luoyang 471003, China

**Keywords:** *Bacillus subtilis*, AFB_1_, Aflatoxin-degrading enzyme, CAZymes, secondary metabolites, whole genome sequence

## Abstract

**Simple Summary:**

AFB_1_ is the most toxic mycotoxin known and is considered a class of carcinogens with serious damaging effects on various tissues and organs, especially the liver. In this study, we screened a *Bacillus subtilis* ZJ20 that effectively degraded AFB_1_ (84.23% degradation rate) and sequenced and analyzed its complete genome. The general traits of *Bacillus subtilis* ZJ20 and its genome were outlined, and further genetic analyses showed that the strain had strong enzyme content, metabolic activity, and antioxidant capacity. Potential antibacterial, antifungal, and antiviral capacities of ZJ20 were hypothesized by metabolic prediction. In addition, genes encoding AFB_1_-degrading enzymes, including chitinase, laccase, lactonase, and manganese peroxidase, were identified in the whole genome of ZJ20, demonstrating that ZJ20 degrades AFB_1_ through multiple enzymes.

**Abstract:**

As a class I carcinogen, aflatoxin can cause serious damage to various tissues and organs through oxidative stress injuries. The liver, as the target organ of AFB_1_, is the most seriously damaged. Biological methods are commonly used to degrade AFB_1_. In our study, the aflatoxin B_1_-degrading strain ZJ20 was screened from AFB_1_-contaminated feed and soil, and the degradation of AFB_1_ by ZJ20 was investigated. The whole genome of strain ZJ20 was analyzed, revealing the genomic complexity of strain ZJ20. The 16S rRNA analysis of strain ZJ20 showed 100% identity to *Bacillus subtilis IAM 12118*. Through whole gene functional annotation, it was determined that ZJ20 has high antioxidant activity and enzymatic activity; more than 100 CAZymes and 11 gene clusters are involved in the production of secondary metabolites with antimicrobial properties. In addition, *B. subtilis* ZJ20 was predicted to contain a cluster of genes encoding AFB_1_-degrading enzymes, including chitinase, laccase, lactonase, and manganese oxidase. The comprehensive analysis of *B. subtilis* provides a theoretical basis for the subsequent development of the biological functions of ZJ20 and the combinatorial enzyme degradation of AFB_1_.

## 1. Introduction

Aflatoxin (AFT) is the most stable and toxic of the known mycotoxins, and it is a bifuranocyclic toxin produced by certain strains of *Aspergillus flavus* and *Aspergillus parasiticus* [1,2]. There are more than 20 aflatoxin derivatives, of which aflatoxinB_1_ (AFB_1_) is the most common and most toxic and is classified as a group I carcinogen due to its strongest carcinogenic and mutagenic properties [3]. Aflatoxins are widely present in agricultural products such as legumes and grains, and they reach animal tissues, milk, and eggs through a “carry-over” mechanism after feeding animals with contaminated feedstuffs [4]. If humans eat food containing AFB_1_, it can cause damage to various organs such as the gastrointestinal tract [5], nervous system [6], lungs [7], and liver [8]. The liver is the most affected by aflatoxins and can be damaged by oxidative stress, leading to impaired liver cell function or apoptosis [9].

Genomics, proteomics, transcriptomics, metabolomics, and other omics techniques are widely used in the study of bacteria, toxins, and viruses. Genomics is the study of all the genes of an organism. Genomics allows the identification of relevant genes that play a role in the degradation of mycotoxins by bacteria and the analysis of the structure, function, evolution, localization, editing, and role of the relevant genomes [10]. Genomics includes functional genomics, structural genomics, epigenomics, and macro-genomics. Different genomic analyses allow for describing the functions and interactions of genes and proteins, the three-dimensional structure of proteins, and epigenetically modified genetic material [11]. Genome-wide analysis plays an important role in discovering gene functions and better understanding the biological functions, pathogenic mechanisms, toxin degradation mechanisms, and beneficial mechanisms of bacteria. For example, genome-wide analysis can reveal the active enzyme of *Bacillus licheniformis BL-0101* that acts on AFB_1_ [12]. In addition, the mechanisms by which food-grade bacteria function in the gut were revealed by postgenomics, comparative genomics, and other analyses [13].

Microbial degradation of AFB_1_ is mostly through the destruction of toxic sites or biotransformation, and enzymes, as the substances that play a major catalytic role in the study of AFB_1_ degradation mechanisms, occupy an important position in the biodegradation of AFB_1_. Enzymes with the ability to degrade AFB_1_ toxin have been reported to include laccases [14], manganese peroxidase [15], dye-decolorizing peroxidase [16], and lactonase [17], as well as chitinase, which is antagonistic to *Aspergillus flavus* strains. These degradation enzymes act on *Aspergillus flavus* strains and aflatoxins through different degradation mechanisms. Chitinases degrade the chitin that makes up the fungal cell wall and inhibit the growth of fungal strains [18], oxidases and peroxidases oxidize AFB_1_ to non-toxic or less toxic products through oxidation reactions, and lactases open the inner lipid ring structure to degrade AFB_1_ [19].

## 2. Materials and Methods

### 2.1. Screening of AFB_1_-Degrading Strains

Five grams of moldy feed was mixed with 100 mL of sterile water; 100 μL of the upper suspension was taken after 20 min of resting, diluted to 10^−1^, 10^−2^, 10^−3^, 10^−4^, and 10^−5^ with a gradient of sterile water, and inoculated with 2% in liquid medium containing coumarin.

After incubation in a shaker at 37 °C for 24 h, the bacterial solution was taken and coated in solid medium containing coumarin, and single colonies were selected for purification and preserved for primary screening of bacteria.

The bacteria from the initial screening were inoculated in Lysogeny broth (LB) liquid medium containing 0.5 μg/mL AFB_1_, incubated at 37 °C for 48 h, and then sampled to determine the content of AFB_1_ for bacterial re-screening.

### 2.2. Detection of AFB_1_ Degradation Ability

The content of AFB_1_ was determined by high-performance liquid chromatography (HPLC). The supernatant was mixed with acetonitrile and water (acetonitrile/water = 84:16) and 1 mL of n-hexane, shaken, and left for 1 h. Then, 200 μL of AFB_1_ degradation solution was taken through a 0.22 filter membrane and added to the injection bottle for determination.

Chromatographic conditions: Waters XBridge C18 (5 μm 4.6 × 250 mm) column. Mobile phase: phase A: water; phase B: acetonitrile–methanol solution (50 + 50). Gradient elution: 24%B (0–6 min), 35%B (8.0–10.0 min), 100%B (10.2–11.2 min), 24%B (11.5–13.0 min). Flow rate: 1.0 mL/min; column temperature: 40 °C; injection volume: 50 μL. Detection wavelength: excitation wavelength 360 nm; emission wavelength 440 nm.

### 2.3. General Characteristics of ZJ20

The ZJ20 strain was inoculated in LB solid medium, and the morphological characteristics of single colonies were observed. Single colonies were selected and inoculated in the liquid medium, and the morphological characteristics of the strain were observed under the microscope by taking the Gram stain of the bacterial solution.

### 2.4. Genome Sequence Determination and Assembly of Strain ZJ20

Individual colonies of strain ZJ20 were inoculated in an LB solid medium and incubated at 37 °C in an incubator shaker for 12 h. The whole genome of ZJ20 was extracted using the Bacterial Genomic DNA Extraction Kit DP302 (TIANGEN Biochemical Technology, Beijing, China) according to the manufacturer’s instructions. The DNA was stored at −20 °C for further use.

Whole genome sequencing of ZJ20 was performed by Sangon Biotech Co., Ltd. (Shanghai, China). Sequencing was performed using the Illumina platform [20], and after accounting for information such as raw data quality, the sequencing data quality of the samples was assessed visually using FastQC 0.11.2 [21].

The sequencing data source was spliced using SPAdes 3.5.0 [22] after filtering for fragment reads (less than 35 nt) that contained spliced, low-quality, and segmented fragments. After splicing, GapFiller 1.11 [23] was used to complement the GAP of the contig obtained by splicing, and finally, sequence correction was performed with PrlnSeS-G (primer-initiated sequence synthesis for genomes) 1.0.0 [24].

Gene, tRNA, rRNA, and other gene elements were predicted with Prokka1.10 [25].

### 2.5. Molecular Confirmation of B. subtilis ZJ20

The macro-genome of ZJ20 was compared with the Non-Redundant (NR) database in the National Center for Biotechnology Information (NCBI) to check the proximity of the species transcript sequences to similar species and the functional information of homologous sequences. The spliced sequence files were compared with the data in the NCBI 16S database for Blast comparison analysis. Twelve bacterial strains with high homology were selected, and a phylogenetic tree was constructed using MEGA 11.0.13 [26] software, and the Maximum Likelihood method was used for phylogenetic analysis.

### 2.6. Functional Annotation of the ZJ20 Genome

The whole genome of ZJ20 was annotated using the Prokka online site (https://proksee.ca/projects/new accessed on 1 March 2023) and a circular map of the strain ZJ20 genome was constructed using CGviewer (https://paulstothard.github.io/cgview/ accessed on 1 March 2023) [27].

The protein sequence of the ZJ20 gene was compared with Clusters of Orthologous Genes (COG) [28], Swissprot, and TrEMBL databases using NCBI Blast+ [29] to obtain the corresponding functional annotation information. The Gene Ontology (GO) functional annotation information was obtained based on the results of gene annotation with Swissprot and TrEMBL.

Finally, the annotation information for the gene Kyoto Encyclopedia of Genes and Genomes (KEGG) [30] was obtained using KAAS (KEGG Automatic Annotation Server) [31].

### 2.7. Prediction of Genes Encoding for CAZymes and Secondary Metabolites in B. subtilis ZJ20

The predicted B. subtilis ZJ20 protein sequence was aligned with the carbohydrate-active enzymes (CAZymes) [32] database using the dbCAN [33] online annotation platform (https://bcb.unl.edu/dbCAN2/index.php accessed on 14 February 2023) to predict the gene encoding the carbohydrate-active enzyme in ZJ20.

Identification of secondary metabolite gene clusters using the antiSMASH (antibiotics and Secondary Metabolite Analysis Shell) [34] online annotation site (http://antismash.secondarymetabolites.org/ accessed on 6 March 2023).

### 2.8. Mining of Potential Aflatoxin AFB_1_-Degrading Enzymes of B. subtilis ZJ20

The coding sequences of reported AFB_1_-degrading enzymes in bacteria were downloaded from NCBI, and similar sequences of the downloaded sequences were screened in the whole genome of ZJ20 using TBtools 1.115 [35] software. Then, multiple sequence comparisons were performed using the ClustalW tool of MEGA 11.0.13 software.

The aligned sequences were used for protein secondary structure prediction and the phylogenetic tree of the enzyme using Jalview 2.11.2.6 [36] software. Protein tertiary structure prediction is performed through the online site SWISS-MODEL [37] (https://swissmodel.expasy.org accessed on 20 March 2023).

The experimental flow is shown in Figure 1.

## 3. Results

### 3.1. Screening of AFB_1_-Degrading Strains

The bacteria in the moldy feed were enriched and cultured, and 79 strains of bacteria were isolated. These 79 strains were subjected to primary screening in a primary screening medium containing coumarin as the only carbon source, and 12 strains with stable growth and good condition were obtained. The 12 strains were numbered as ZJ3, ZJ10, ZJ11, ZJ16, ZJ6, ZJ13, ZJ20, ZJ27, ZJ31, ZJ32, ZJ8, ZJ22.

The 12 strains obtained from the initial screening were co-cultured with LB medium containing 0.5 μg/mL for 48 h. The residual amount of AFB_1_ in the medium was detected by HPLC and the degradation efficiency of the strains was calculated, and the results are shown in Table 1. The results showed that the degradation efficiency of strain ZJ20 on AFB_1_ was the highest, which could reach 84.23%.

### 3.2. Bacillus Identification

Strain ZJ20 is a Gram-positive bacillus that grows fast and often forms a white film on the liquid surface in an LB liquid medium. The colonies in the solid medium are larger, white, and irregular in shape, with smooth depressions in the middle and ruffled elevations all around.

Gram staining microscopy for blue-purple rods, uniform coloring, and no pods can form endophytic resistant bacilli, located in the center of the body or slightly off (Figure 2).

The genome of the *B. subtilis* ZJ20 strain was compared with the NR database to make a pie chart of the homology distribution of ZJ20. Each sector in the chart represents one species, and the larger the area of the sector, the greater the number of sequences compared to that species. As shown in Figure 3, strain ZJ20 has high homology with *Bacillus* spp. and *Bacillus substilis* species.

The 16sRNA sequence of strain ZJ20 was 1454 bp in length, and the sequence was submitted to GenBank with the accession number OR453222. The complete 16S rRNA of ZJ20 was compared to Blast, and the comparison showed that *B. subtilis* ZJ20 was the closest to the *Bacillus subtilis* strain (more than 99% identity), which was the same as the result of the NR database comparison.

Four *Bacillus subtilis* strains and four other *Bacillus* spp. homologous ZJ20 strains were selected to construct a bacterial phylogenetic tree. As shown in Figure 4, strain ZJ20 was in the same branch as *Bacillus subtilis* strain *IAM12118* and had the highest homology. The ZJ20 strain can be identified in *Bacillus subtilis* species (*Bacillus subtilis*), named *Bacillus subtilis* ZJ20, abbreviated as *B. subtilis* ZJ20.

### 3.3. Basic Characteristics of the Genome

The genomic statistics of *B. subtilis* ZJ20 are shown in Table 2. The total length of all coding genes was 4,326,340 bp, and the length of coding genes ranged from 45 to 10,806 bp, with an average length of 817.69 bp. Most of the genes were between 200 and 1000 bp (Figure 5), with an average G + C content of 42.9%.

The *B. subtilis* ZJ20 genome contains 11 copies of rRNA manipulators (16S, 23S, and 5S RNAs), 85 tRNA genes, and 4659 coding sequences (CDS), of which 4432 are functionally annotated. This Whole Genome Shotgun project has been deposited at GenBank under the accession JAVFVM000000000. The version described in this paper is version JAVFVM010000000.

### 3.4. COG Annotation Results

All protein-coding genes were annotated with the protein direct homology cluster database (COG). The functional categories defined according to the COGs showed that *B. subtilis* ZJ20 contains a large number of proteins involved in amino acid transport and metabolism (COG E) and transcription (COG K), followed by carbohydrate transport and metabolism (COG G) (Table 3). The annotated gene circle map is shown in Figure 6.

### 3.5. GO Annotation Results

The gene sequences encoding the protein encoded by *B. subtilis* ZJ20 were aligned with the Gene Ontology (GO) database to obtain functional annotation information for the protein encoded by *B. subtilis* ZJ20. The proteins of *B. subtilis* ZJ20 are divided into three main categories: Cellular Component (CC); Biological Process (BP); and Molecular Function (MF).

ZJ20 has a large number of genes responsible for metabolic processes, cellular processes, cells, binding, and catalytic activity, followed by more genes coding for antioxidant activity and enzyme regulator activity (Figure 7). This suggests that *B. subtilis* ZJ20 may have high enzyme content, metabolic activity, and antioxidant capacity.

### 3.6. KEGG Annotation Results

The whole genome of *B. subtilis* ZJ20 was compared with the KEGG database, and most of the genes in all KEGG pathways were involved in carbohydrate metabolism, membrane transport, and amino acid metabolism (Figure 8). Most of the genes in amino acid metabolism are related to cysteine, methionine, phenylalanine, D-alanine, histidine, lysine, complexion, arginine, proline, tryptophan metabolism, and the synthesis of glycine, valine, and lysine.

In carbohydrate metabolism, most genes are involved in the synthesis of pantothenic acid, CoA, fatty acids, unsaturated fatty acids, folic acid, and peptidoglycans; and fatty acid, biotin, propionic acid, starch, sucrose, glutathione, fructose, galactose, mannose, purine, glyoxylate, and dicarboxylic acid metabolism. There are also genes involved in biological pathways such as the citric acid cycle (TCA cycle), ABC transporter, tetracycline biosynthesis, and glycolysis (Appendix A).

It shows that *B. subtilis* ZJ20 has strong biochemical metabolism, membrane permeability, and reproduction ability, which provide the basis for the production of a large number of functional enzymes.

### 3.7. CAZy Database Annotation Result

CAZymes are carbohydrate-active enzymes that play a key role in carbohydrates. In the CAZy database, CAZymes are classified into five categories, namely GH (glycoside hydrolase), GT (glycosyltransferase), CE (carbohydrate esterase), PL (polysaccharide lyase), and AA (auxiliary active).

Sequence analysis of the ZJ20 genome identified 127 CAZymes family enzymes, including 43 GH, 42 GT, 11 CBM, 11 CE, 7 PL, and 2 AA. In addition, 11 enzymes of the CBM/GH complex type were identified (Figure 9). Glycoside hydrolases and glycosyltransferases had the highest number of genes, accounting for 33.9% and 33.1% of the total number of CAZymes genes, respectively.

### 3.8. Prediction of Secondary Metabolites of B. subtilis ZJ20

The obtained whole genome sequence of ZJ20 was compared to the secondary metabolite database, and a variety of secondary metabolites encoding antibacterial properties were identified in *B. subtilis* ZJ20.

It contains five gene clusters encoding NRPS (non-ribosomal peptide synthase); two gene clusters encoding terpenoids; and two gene clusters encoding class I lactose peptides, one of which shows 100% similarity to genes involved in the synthesis of chymotrypsin. There was also one gene cluster each encoding T3PKS (class III polyketide synthase), CDPS (cyclic dipeptide synthase), and bacillus lysozyme; one gene cluster encoding a cecropeptide class showed 100% similarity to Subtilosin A. Among the five gene clusters encoding NRPS, two gene clusters showed 100% similarity to fengycins and bacillibactin, respectively (Figure 10).

### 3.9. Mining Potential AFT-Degrading Enzymes of Strain B. subtilis ZJ20

In addition to the chitinase with possible AFB_1_ degradation potential identified in the above analysis of carbohydrase, a laccase, two lactonases, and four protein fragments of manganese peroxidase were identified in ZJ20.

The laccase sequence in ZJ20 was compared with three segments of laccase protein sequences for multiple sequence comparison as well as secondary structure prediction (Figure 11). In addition, the tertiary structure of the protein was also very similar to the known tertiary structure of laccase (Figure 12), indicating that the laccase produced by ZJ20 has the potential to degrade AFB_1_.

ZJ20 contains the protein sequences of two lactonases, and these two protein sequences were used to construct a phylogenetic tree together with six lactonases of different origins (Figure 13). The results indicated that PROKKA03497 was distantly related to other lactonases and that it was a new lactonase to be investigated.

Of the four manganese peroxidase protein fragments, one had 100% sequence identity with the identified manganese peroxidase coding sequence. As shown in Figure 14, five different manganese peroxidase protein sequences were selected to construct a phylogenetic tree with the four sequences in ZJ20 (Figure 15). The results showed that PROKKA04540 was distantly related to the other proteins, and the protein might belong to a different family from other proteins, which has a greater potential for exploitation.

## 4. Discussion

Aflatoxins are natural carcinogens with multiple toxicities and are present in moldy feed and cereals, with AFB_1_ being a class I carcinogen that can damage a variety of tissues and organs of the body. A variety of enzymes have been reported to degrade aflatoxin, and the focus of current research is to develop an efficient degrading enzyme for aflatoxin to reduce the risk of aflatoxin to humans and animals.

In our study, *B. subtilis* ZJ20 showed good degradation of AFB_1_ and produced effects through extracellular enzymes. *B. subtilis* ZJ20 encoded a total of four AFB_1_-degrading enzymes, including chitinase, laccase, lactonase, and manganese peroxidase. AFB_1_ was degraded by a combination of multiple enzymes to achieve higher degradation efficiency.

Thereafter, whole genome sequencing was combined with bioinformatics analysis to provide more information to explore more biological functions of strain ZJ20.

Amino acids are small molecules that constitute proteins and play an important role in protein synthesis, maintaining normal body metabolism, improving body immunity, and increasing energy [38]. ABC transporter proteins are a large and functionally diverse family of transmembrane transport proteins responsible for the transmembrane transport of a variety of substances that are essential for cellular detoxification, membrane homeostasis, and lipid transport [39]. Fatty acids maintain normal cellular physiological functions, maintain the relative fluidity of cell membranes, and provide energy to cells. And some fatty acids, short-chain fatty acids (SCFAs), are involved in the immune system, regulating intestinal pH, intestinal permeability, intestinal microbiota, and reducing intestinal inflammation [40]. Biotin, a member of the B vitamin family, enhances the host’s immune response, resists infection, and maintains normal growth and development [41]. Tetracycline, as a broad-spectrum antibacterial agent, has a good inhibitory effect on both Gram-positive and Gram-negative bacteria [42]. In our study, we found that *B. subtilis* ZJ20 contains genes related to amino acids, fatty acid metabolism and synthesis, biotin synthesis, and tetracycline biosynthesis. This suggests that *B. subtilis* ZJ20 can exert beneficial effects on animals as a probiotic by controlling the metabolism of amino acids and fatty acids, enhancing the digestion and absorption of nutrients in animals, and inhibiting the growth of intestinal pathogenic bacteria.

Carbohydrates are essential for the survival of microorganisms, acting not only as a carbon source for bacteria but also as a means of attachment to the host and as a barrier to host infection [43]. Glycoside hydrolases play an important role in the synthesis of oligosaccharides, the synthesis of alkyl and aromatic glycosides, the glycosylation of amino acids and peptides, and the glycosylation of antibiotics. Glycosyltransferases use activated sugar donors to drive glycosyl transfer to appropriate acceptors, and the formation of glycosidic bonds affects the metabolic choreography of small molecules and the formation of numerous glycolipids, glycopeptides, and lipopolysaccharides [44].

The more numerous CAZymes gene families in *B. subtilis* ZJ20 were 5 GH23, 4 GH32, 17 GT2, and 10 GT4 family genes, respectively (Appendix A). The GH23 family of glycoside hydrolases mainly includes lysozyme type G, chitinase, and peptidoglycan lyase, which can degrade fungal and pathogenic cell walls and thus inhibit pathogenic growth. Chitinase also has the potential to degrade fungal toxins such as AFB_1_. In addition, the β-glucosidases GH1 and GH4, xyloglucanase GH16, and endoglucanase GH51 gene families also have potential antifungal activity, and fructosyltransferase GH32 can also stimulate the growth of intestinal probiotics [45].

GT4, as the largest class of the GT family, contains some enzymes that have potential therapeutic significance. GT4, along with GT2, as the ancestral reversal and retention family of the GT family, can modulate antibiotic activity by glycosylating antibiotics and changing the location, type, and amount of sugar [44]. The binding properties of CBM improve the catalytic function of CAZymes by targeting the enzyme to the substrate, increasing substrate–enzyme proximity, and disrupting the crystallinity of the insoluble substrate fraction [46]. Carbohydrate-active enzymes can break down complex carbohydrates such as cellulose, starch, glycogen, and polysaccharides into small-molecule compounds, which facilitate nutrient absorption from the intestinal epithelium. These genes encoding CAZymes allow *B. subtilis* ZJ20 to produce carbohydrate-active enzymes that synthesize carbohydrates and metabolize sugars and have antifungal, pathogen-inhibiting, antibiotic-activating, and nutrient-absorbing effects.

In addition, genes encoding NRPS, terpenes, class I lactose peptides, T3PKS, CDPS, bacillus lysozyme, and sequestered peptides were identified in the analysis of secondary metabolites. Various secondary metabolites encoded by ZJ20 have been reported to have antifungal [47,48], broad-spectrum antibacterial [49,50], antiviral [51], and antioxidant [52] effects, and some metabolites have also been predicted to have immunomodulatory, antitumor, and antiparasitic effects [53,54], laying the groundwork for the excavation of other biological effects of ZJ20. Among the secondary metabolites encoded by ZJ20, there is also a metal carrier that exhibits 100% similarity to the iron carrier and can chelate Fe^3+^, Al^3+^, Pb^2+^, Cu^2+^, and other metal ions bound [55] to bioremediate metal-contaminated soil. *B. subtilis* ZJ20 has seen large developments not only in the farming and pharmaceutical industries, but also in environmental control and plant biocontrol. The potential of *B. subtilis* ZJ20 is not only in the farming and pharmaceutical industries but also in environmental control and plant biocontrol.

Overall, *B. subtilis* ZJ20 can function as a feed additive, plant and environmental biocontrol bacterium, and bioengineering bacterium due to its ability to produce a variety of enzymes, antibiotics, and other biologically active substances. ZJ20 can effectively degrade AFB_1_, so when it is added as a probiotic to the feed, it can not only alleviate the effect of AFB_1_ in moldy feed on the animal’s body, but also improve the animal’s intestinal flora balance.

## 5. Conclusions

In summary, our screened strain, ZJ20, could effectively degrade aflatoxin B_1_ with a degradation rate of 84.23%. ZJ20 was identified as *Bacillus subtilis* according to the general morphological characteristics of the strain and 16Sr RNA comparison. After sequencing and assembling the whole bacterial genome, we understood the basic genomic features of ZJ20 and functionally annotated its protein-coding genes.

We identified genes involved in amino acid metabolism, lipid metabolism, and vitamin metabolism and synthesis in a genome-wide KEGG analysis of *B. subtilis* ZJ20, confirming the probiotic properties of ZJ20. Active enzymes with inhibitory effects on fungi and bacteria were identified in CAZymes and secondary metabolite analyses, and four potential AFB_1_-degrading enzymes were found in *B. subtilis* ZJ20 by multiple sequence comparison. The comprehensive analysis of ZJ20 in this paper lays a theoretical foundation for the subsequent development of more biological functions of ZJ20 and the combined degradation of AFB_1_ by multiple biological enzymes.

For future research, on the one hand, we can continue to explore the biological functions of *B. subtilis* ZJ20 as a probiotic so that it can perform the corresponding probiotic functions. On the other hand, the cloning of AFB_1_-degrading enzymes to degrade AFB_1_ more effectively is a very worthwhile research direction.

## Figures and Tables

**Figure 1 biology-12-01195-f001:**
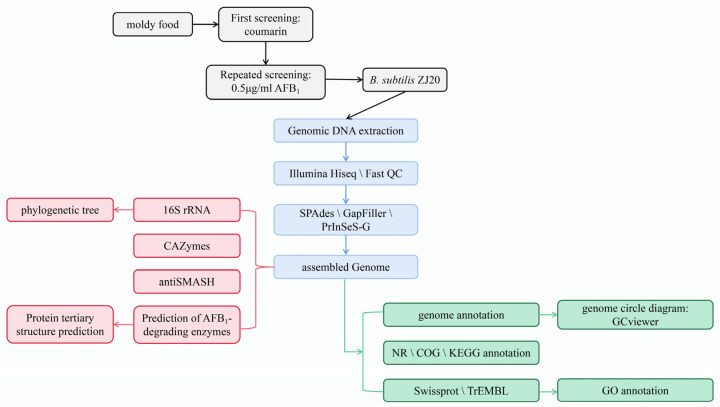
Experimental technology roadmap. (Black: strain screening; blue: genome sequencing and assembly; red: bacterial product prediction; green: genome annotation.)

**Figure 2 biology-12-01195-f002:**
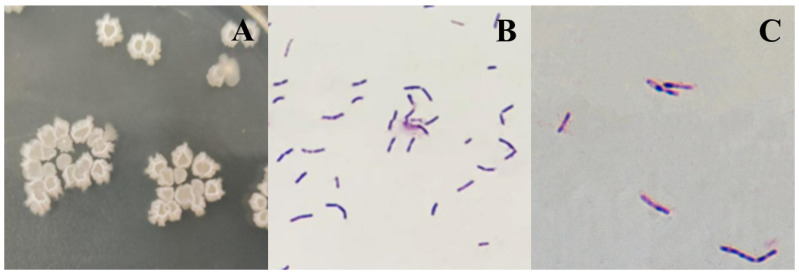
General characteristics of strain ZJ20. (**A**) Colony morphology. (**B**) Gram stain microscopy (1000×). (**C**) Microscopic examination of the Gram stain of budding spores (1000×).

**Figure 3 biology-12-01195-f003:**
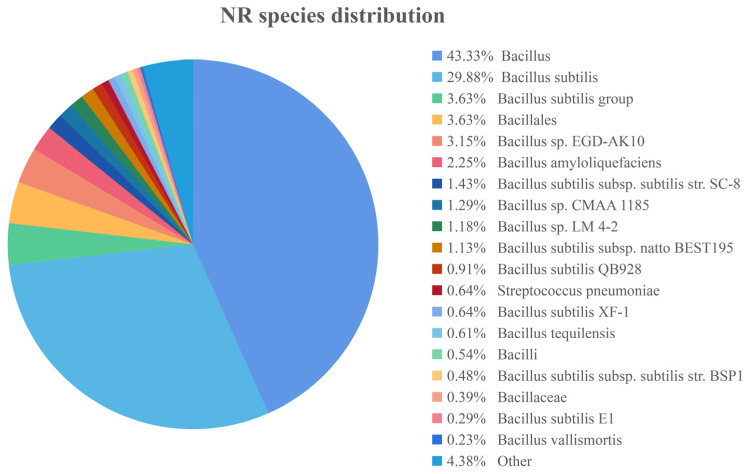
Pie chart of the homology distribution of *B. subtilis* ZJ20.

**Figure 4 biology-12-01195-f004:**
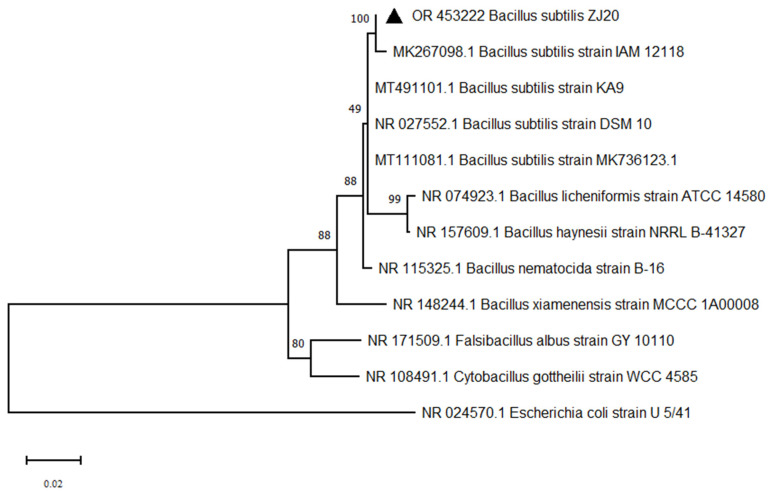
Phylogenetic tree construction based on the full 16S rRNA of *Bacillus subtilis* ZJ20 by the maximum likelihood method.

**Figure 5 biology-12-01195-f005:**
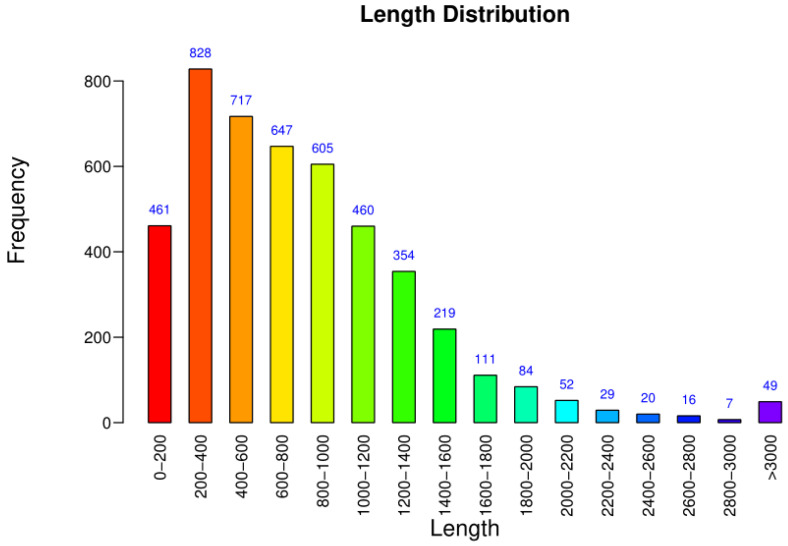
Distribution of coding gene lengths. The horizontal axis represents the length of the protein sequence, and the vertical axis represents the number of genes within the interval.

**Figure 6 biology-12-01195-f006:**
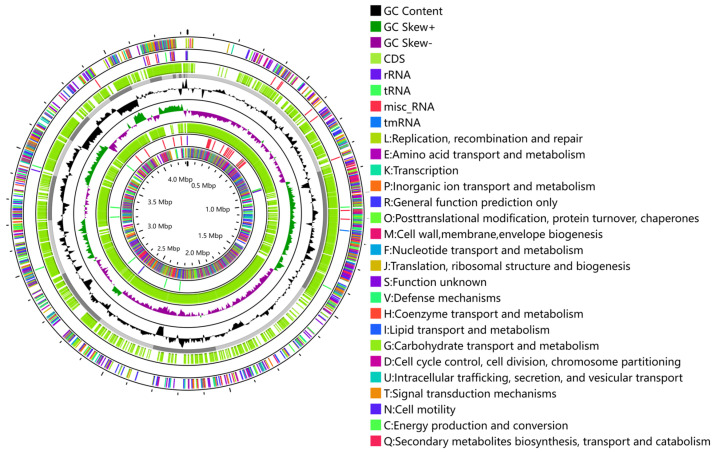
Genome circle map of *B. subtilis* ZJ20. From edge to center: forward strand of protein−coding genes with COG data annotation information; forward strand colored by RNA category; forward strand of protein−coding genes; forward strand colored according to GC content and GC skewness; reverse strand of protein−coding genes; reverse strand colored by RNA category; reverse strand of protein-coding genes with COG data annotation information.

**Figure 7 biology-12-01195-f007:**
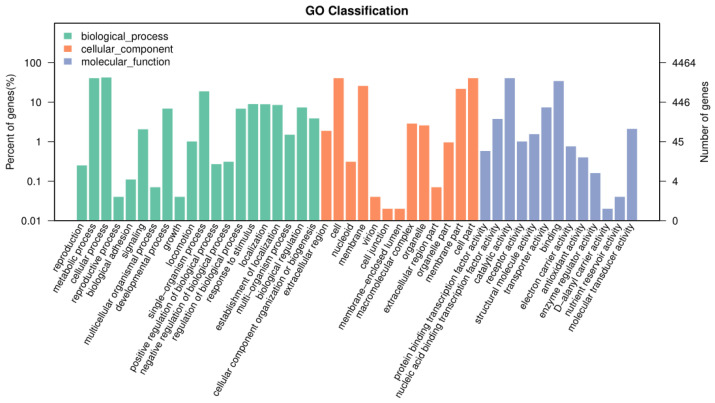
GO annotation results of *B. subtilis* ZJ20.

**Figure 8 biology-12-01195-f008:**
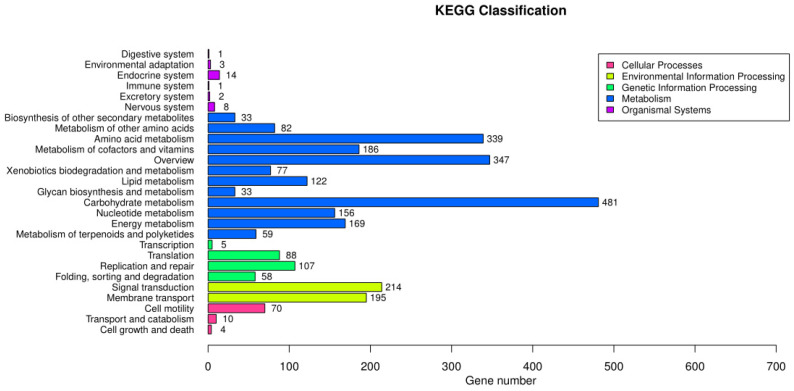
Taxonomic map annotation of the genomic KEGG metabolic pathway of *B. subtilis* ZJ20.

**Figure 9 biology-12-01195-f009:**
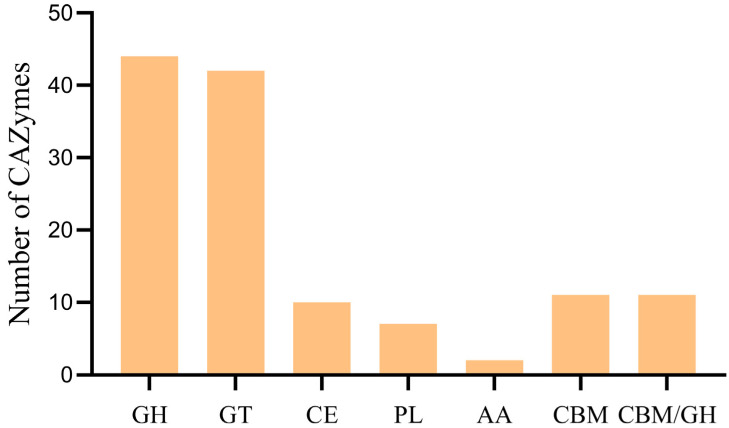
Distribution of the carbohydrate active enzyme (CAZy) family protein identified in the genome of *B. subtilis* ZJ20.

**Figure 10 biology-12-01195-f010:**
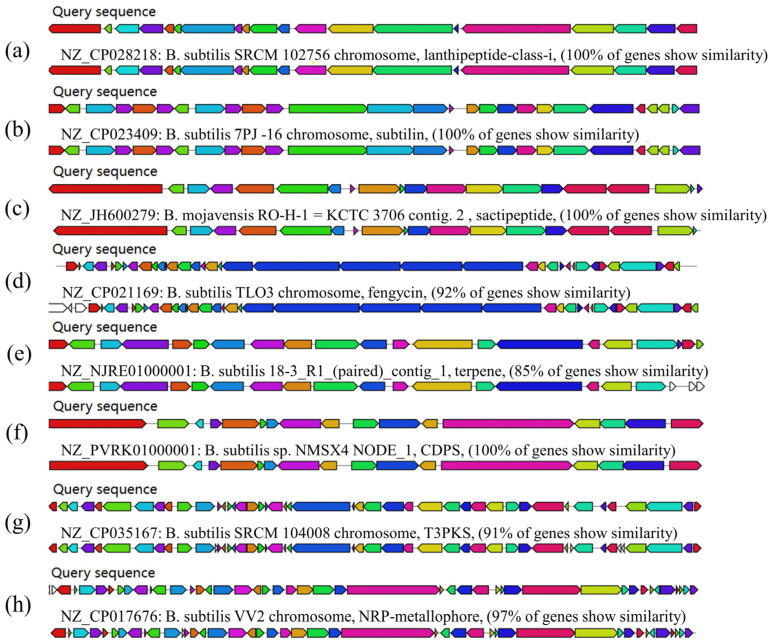
Analysis of secondary metabolites of *B. subtilis* ZJ20: (**a**) lanthipeptide-class-i; (**b**) subtilin; (**c**) sactipeptide; (**d**) fengycin; (**e**) terpene; (**f**) CDPS; (**g**) T3PKS; (**h**) NRP-metallophore.

**Figure 11 biology-12-01195-f011:**
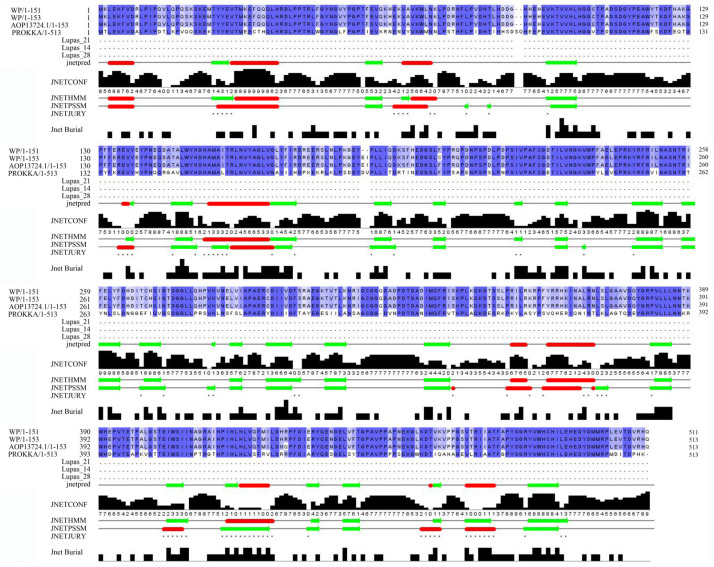
Laccase comparison and secondary structure prediction in the ZJ20 genome: the red bar is α-helix and the green arrow is β-fold. The black color indicates the number of identical amino acids and the blue color indicates the sequence similarity, the darker the color, the higher the sequence similarity.

**Figure 12 biology-12-01195-f012:**
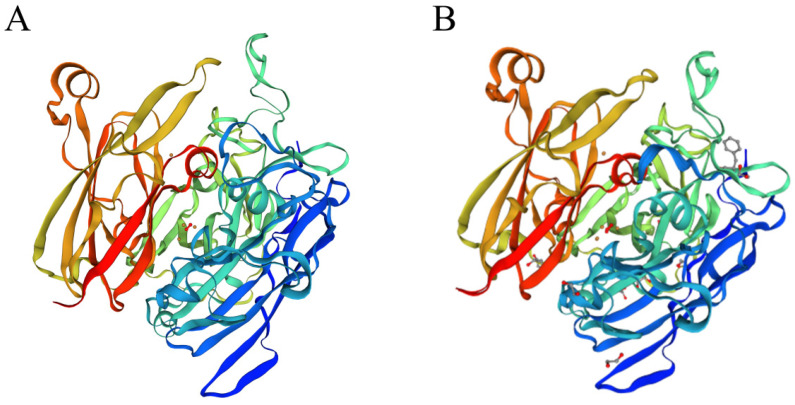
(**A**) Prediction of the laccase tertiary structure in the ZJ20 genome. (**B**) The predicted tertiary structure of laccase has been reported.

**Figure 13 biology-12-01195-f013:**
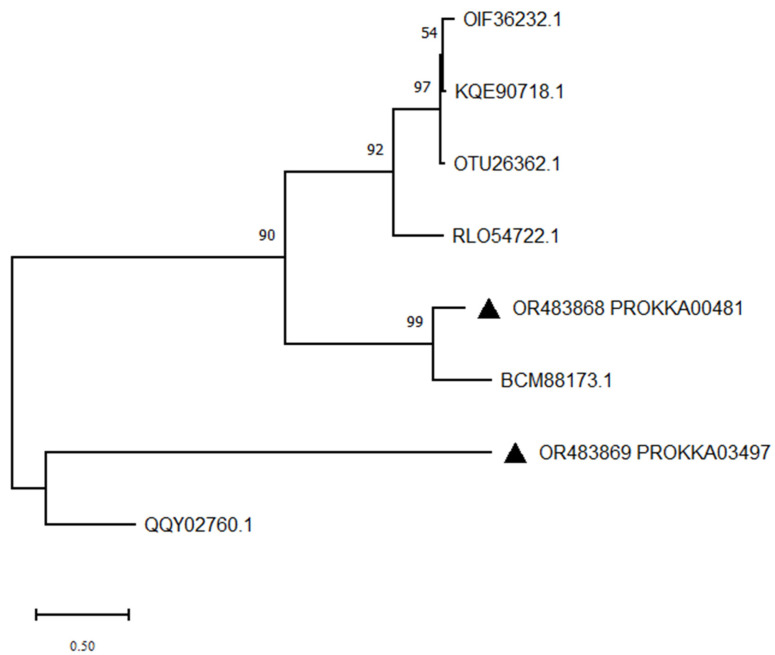
Phylogenetic tree of lactonase protein sequences in ZJ20 constructed by the maximum likelihood method: triangles indicate endostatin protein sequences in ZJ20.

**Figure 14 biology-12-01195-f014:**
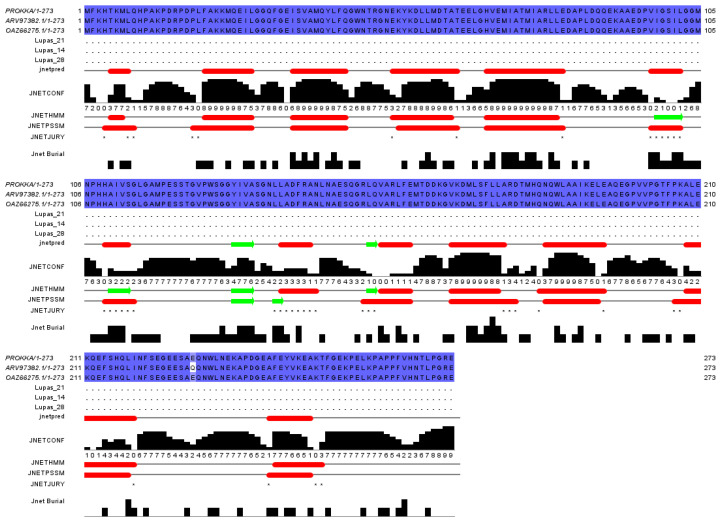
Comparison and secondary structure prediction of the protein sequence PROKKA04370 encoding manganese peroxidase in ZJ20: the red bar is α-helix and the green arrow is β-fold. The black color indicates the number of identical amino acids and the blue color indicates the sequence similarity, the darker the color, the higher the sequence similarity.

**Figure 15 biology-12-01195-f015:**
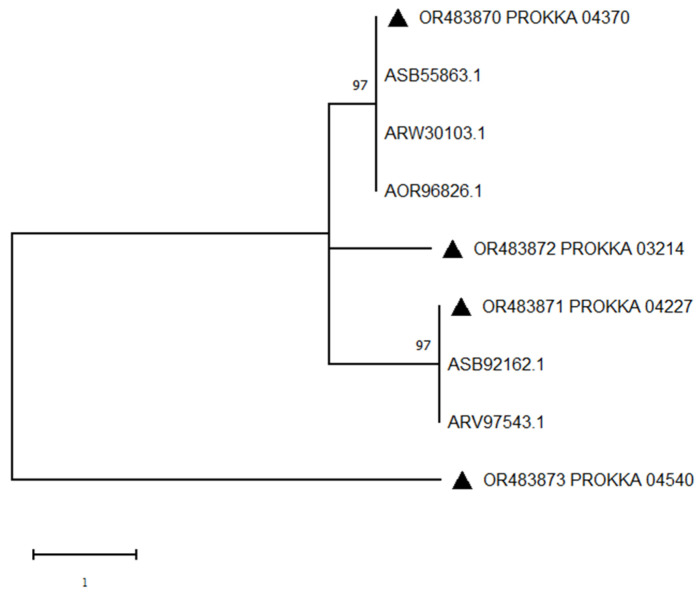
Phylogenetic tree of manganese peroxidase protein sequences in ZJ20 constructed by the maximum likelihood method: triangles indicate the manganese peroxidase protein sequences in ZJ20.

**Table 1 biology-12-01195-t001:** Degradation rate of AFB_1_-degrading strains.

Strain Number	AFB_1_ Degradation Rate (%)	Strain Number	AFB_1_ Degradation Rate (%)
ZJ3	40.26 ± 0.22	ZJ16	45.86 ± 1.36
ZJ6	52.09 ± 1.23	ZJ20	84.23 ± 0.13
ZJ8	42.86 ± 2.16	ZJ22	38.73 ± 0.32
ZJ10	52.06 ± 1.16	ZJ27	54.28 ± 2.03
ZJ11	35.86 ± 1.14	ZJ31	41.49 ± 0.12
ZJ13	68.47 ± 0.18	ZJ32	53.29 ± 1.24

**Table 2 biology-12-01195-t002:** General genome features of *B. subtilis* ZJ20.

Class	Number
Size (base)	4,326,240
G + C content (%)	42.9
Protein Coding Genes	4659
Min length (base)	45
Max length (base)	10,806
Average length (base)	817.69
Total coding gene (base)	3,809,638
Coding ratio (%)	88.06
tRNA	85
rRNA	11
Repeat Region	
Repeat Region Count	0
Total Repeat Region (base)	0
Repeat Ratio (%)	0

**Table 3 biology-12-01195-t003:** COG categories of *B. subtilis* ZJ20.

COG Code	Number	Proportion (%)	Description
C	164	5.35	Energy production and conversion
D	37	1.21	Cell cycle control, cell division, and chromosome partitioning
E	248	8.09	Amino acid transport and metabolism
F	82	2.68	Nucleotide transport and metabolism
G	260	8.48	Carbohydrate transport and metabolism
H	125	4.08	Coenzyme transport and metabolism
I	72	2.35	Lipid transport and metabolism
J	168	5.48	Translation, ribosomal structure, and biogenesis
K	257	8.38	Transcription
L	133	4.34	Replication, recombination, and repair
M	185	6.04	Cell wall/membrane/envelope biogenesis
N	24	0.78	Cell motility
O	91	2.97	Posttranslational modification, protein turnover, chaperones
P	171	5.58	Inorganic ion transport and metabolism
Q	60	1.96	Secondary metabolite biosynthesis, transport, and catabolism
R	400	13.05	General function prediction only
S	356	11.62	Function unknown
T	129	4.21	Signal transduction mechanisms
U	45	1.47	Intracellular trafficking, secretion, and vesicular transport
V	58	1.89	Defense mechanisms

## Data Availability

The whole-genome sequence generated in this study was submitted to GenBank with the accession number JAVFVM000000000.

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
