# Peer review of "A Bacillus subtilis Strain ZJ20 with AFB1 Detoxification Ability: A Comprehensive Analysis"

_biology, 2023, doi:10.3390/biology12091195_

Round 1

Reviewer 1 Report

I would like just encourage the authors to speed up publishing this excellent quality paper and think about extending their approach to other bacteria like those used directly or indirectly in biological pest control, like Xenorhadus and Photorhabdus EPN symbiont (EPB) bacteria. I think it would be a great double perspective in this expansion.

Author Response

Dear Reviewer,

Thanks very much for taking your time to review this manuscript. Thanks again for recognizing this experiment and for your valuable suggestions. We will next use the method in this paper to extend the analysis to other bacteria, and we believe that we will get unexpected gains.

Reviewer 2 Report

The manuscript entitled “Genome-wide sequence analysis and potential aflatoxin-degrading enzyme mining of Bacillus subtilis ZJ20” by Huang et al. aimed to study the strain ZJ20 that effectively degraded AFB1, and then predict the potential aflatoxin-degrading enzymes through whole gene functional annotation. In this study, the authors isolated strains from moldy feed and then screened them through the degradation of AFB1. They found the strain ZJ20 that has the highest degradation efficiency. Next, 16s RNA analysis and characterization of ZJ20 showed that it belongs to Bacillus subtilis, confirmed by whole genome sequencing and phylogenetic analysis. Further, they predicted the functional annotation of potential biochemical enzymes, which provides insights into the potential synthesis of AFB1 degrading system of ZJ20.

Although the study is very intersesting, there are some concerns from the reviewer. 

Minor concerns:

1 The authors should proofread the whole manuscript for the logical and grammar issues. 

Here are some examples, but the reviewer didn’t enumerate all of issues below. Please cross-check the whole manuscript. 

Line 155 “... efficiency of strain ZJ20 on AFB1 was higher,...” should be “highest”. 

Line 156 The authors identified ZJ20 after the homology analysis of genome in “Result 3.3 Bacillus identification”. Before confirmation, the author should not claim ZJ20 as B. subtilis ZJ20 in “Result 3.2”. Please check the issue in the figure 1 legend.

2. Line 172. How similar between ZJ20 and Bacillus spp. and substilis species? “High homology” is not clear. 

3. Line 212, please add the symbol % in the table 3 column “Proportion (%)”

4. Line 228. There is no figure 6B in the manuscript.

5. Did the author submit the whole genome sequence of ZJ20 to the public database? If yes, what is the accession number?

The authors should proofread the whole manuscript for the grammar issues. 

Author Response

Dear Reviewer,

Thanks very much for taking your time to review this manuscript. I really appreciate all your constructive comments and suggestions! We have carefully considered the suggestion of Reviewer and make some changes. We have tried our best to improve and made some changes in the manuscript.

The red part of the manuscript has been revised according to your comments. The revision notes, point-to-point, please see the attachment.

Reviewer 3 Report

This manuscript is a detailed exploration of the biodegradation of AFB1 by Bacillus subtilis ZJ20 and the genomic analysis of this strain. While this manuscript provides valuable insights into the topic, there are a few areas that could benefit from improvement:

- The introduction is well-written but could be more concise. Consider condensing the information about aflatoxins and their effects.

   - It might be helpful to structure your introduction to provide a roadmap for the readers, outlining what you'll cover in the paper.

- Make sure to provide citations for all the claims and statements you make. For example, when discussing the effects of aflatoxins on different organs, provide citations for each claim.

- The methods section provides a detailed explanation of your experimental procedures, which is essential. However, consider separating each step into distinct paragraphs for clarity and ease of reading.

- Some parts of your methods are quite technical. Make sure to explain any technical terms or processes in a way that a broader audience can understand. Consider providing brief explanations or references for methods that might not be commonly known.

- When presenting the results, consider using subsections to clearly distinguish between different findings, such as the degradation ability of ZJ20, its general characteristics, genome sequencing, and so on.

- Consider including figures or tables to visually represent complex data or processes, especially in the "Materials and Methods" section. For instance, a flowchart of the screening process could help readers understand it more easily.

- The discussion could be expanded to elaborate on the significance of your findings. How do the AFB1-degrading capabilities and genomic characteristics of ZJ20 contribute to the field of mycotoxin research? How might this strain be practically applied?

- Summarize your main findings and their implications in a concise conclusion section. Reiterate the potential applications of your work and the avenues it opens for future research.

- Proofread your manuscript carefully for grammar, spelling, and sentence structure errors. This will enhance the professionalism of your work.

- Ensure your manuscript follows the formatting guidelines of the journal “Biology”. This includes font, line spacing, figure formatting, and reference style.

- Remember that the manuscript is a comprehensive document that should guide readers through your research process and findings. Keep the language clear, the structure logical, and the content well-organized to ensure a smooth reading experience.

Revised Title such as Enhancing AFB1 Detoxification Using Bacillus subtilis Strain ZJ20: A Comprehensive Analysis

Rewritten Simple Summary: The results of the study were challenging for readers to comprehend. To address this, a more accessible explanation of the findings has been provided. Additionally, a detailed introduction to AFB1, a mycotoxin with limited familiarity, has been included.

Revised Abstract: This study aims to clarify the presented results by restructuring the abstract to incorporate background information, findings, and conclusions. The investigation focuses on enhancing AFB1 detoxification through the utilization of Bacillus subtilis strain ZJ20. Given the unfamiliarity of AFB1, a comprehensive explanation of its properties and risks is provided.

Line 37-38: Aspergillus flavus and Aspergillus parasiticus should be italicized.

Line 38: AFB1 should be spelled out as "aflatoxin B1 (AFB1)" upon first use.

Line 39: "Group I" should be changed to "group I" in lowercase.

Line 61: Bacillus licheniformis BL-0101 -> Scientific name should be in italics, and the strain information for BL-0101 should be included.

Line 69-70: "Aspergillus flavus" should be italicized.

Line 77: Ensure a space between the number and unit. For example, "5 g" and "100 mL."

Line 97: "LB" should be spelled out as "Luria-Bertani (LB)."

Line 100: "gram" should be in lowercase as "gram-negative."

Line 104: Ensure that the city and country names are provided for reagents, kits, and equipment. Check for errors.

Line 107-115: Elaborate on the genome sequencing and bioinformatic analyses performed for strain ZJ20. Provide the versions of software programs used, along with corresponding references.

Line 117: "NR" and "NCBI" should be spelled out as "National Center for Biotechnology Information (NCBI)" and "Non-Redundant (NR)," respectively.

Line 121: Provide a reference for MEGA software.

Line 122: Mention that the phylogenetic tree should be constructed using the maximum likelihood method instead of the neighbor-joining method.

Line 123-130: Spell out abbreviations and provide references for the software used.

Rewrite the poorly written "Materials and Methods" section to improve clarity.

Line 159: "Gram-positive bacillus" should be in lowercase.

Line 166: Insert a space between "(B)" and "Gram."

Line 172: "Bacillus" should be italicized.

Line 175: Instead of the submission number, provide the accession number.

Elaborate on the legend for Figure 2 to provide a detailed understanding. Ensure that the pie chart reflects the proportions of each category.

Regenerate the phylogenetic tree in Figure 3 using the maximum likelihood method. Include the full name of strain ZJ20 along with its accession number in the tree.

Revise the legend for Figure 3 to offer a comprehensive description of the content.

Consider moving Figure 5 to supplementary data and provide an enlarged image.

Increase the font sizes in Figure 10 for better visibility.

Rephrase the legend for Figure 11, ensuring a space between "(A)" and "Prediction."

Regenerate the phylogenetic trees in Figures 12 and 14 using the maximum likelihood method. Include the names of identified genes along with their accession numbers in the trees.

Enhance the quality of Figure 13 for better clarity.

There are numerous revisions required in this manuscript. As it stands, the manuscript is not up to the acceptable standard. I would suggest considering this manuscript for encouragement to resubmit after addressing the revisions.

Moderate editing of English language required.

Author Response

(The authors gave the same response as above.)

Round 2

Reviewer 3 Report

The authors have diligently addressed the reviewer's comments and revised their manuscript accordingly. Based on the thorough revisions made, I wholeheartedly recommend this manuscript for publication in its current form.